# Going Beyond Panaceas: The Diversity of Land Observatory Forms in Africa

**Quentin Grislain** [1,2,3,4,*], **Jeremy Bourgoin** [1,2] , **Ward Anseeuw** [2,5,6], **Perrine Burnod** [1,7], **Eva Hershaw** [6] **and Djibril Diop** [8]

1   UMR TETIS, CIRAD, F-34398 Montpellier, France; jeremy.bourgoin@cirad.fr (J.B.); perrine.burnod@cirad.fr (P.B.)
2   Land Matrix Initiative, 00010 Rome, Italy; w.anseeuw@landcoalition.org
3   UMR 8586—PRODIG, F-93322 Aubervilliers, France
4   Université Paris 1 Panthéon-Sorbonne, F-75231 Paris, France
5   UMR ART-DEV, CIRAD, F-34398 Montpellier, France
6   International Land Coalition, 00010 Rome, Italy; e.hershaw@landcoalition.org
7   Madagascar Land Observatory, 101 Antananarivo, Madagascar
8   Institut Sénégalais de Recherches Agricoles, Bureau d'Analyses Macro-Economiques, BP 3120 Dakar, Senegal; djibrildiopsn@gmail.com
*   Correspondence: quentin.grislain@cirad.fr

**Abstract:** In recent decades, mechanisms for observation and information production have proliferated in an attempt to meet the growing needs of stakeholders to access dynamic data for the purposes of informed decision-making. In the land sector, a growing number of land observatories are producing data and ensuring its transparency. We hypothesize that these structures are being developed in response to the need for information and knowledge, a need that is being driven by the scale and diversity of land issues. Based on the results of a study conducted on land observatories in Africa, this paper presents existing and past land observatories on the continent and proposes to assess their diversity through an analysis of core dimensions identified in the literature. The analytical framework was implemented through i) an analysis of existing literature on land observatories, ii) detailed assessments of land observatories based on semi-open interviews conducted via video conferencing, iii) fieldwork and visits to several observatories, and iv) participant observation through direct engagement and work at land observatories. We emphasize that the analytical framework presented here can be used as a tool by land observatories to undertake ex-post self-evaluations that take the observatory's trajectory into account, or in the case of proposed new land observatories, to undertake ex-ante analyses and design the pathway towards the intended observatory.

**Keywords:** land observatories; diversity; sustainability; land governance; Africa

---

## 1. Introduction

In recent decades, observatories have proliferated [1]. The term "observatory" originally referred to an initiative dedicated to the understanding and prediction of a physical phenomenon [2]. By extension, the term started being used for observation systems often created by State institutions or local authorities to monitor the evolution of an economic or social phenomenon (e.g., national observatory of poverty and exclusion, observatory of racism, national observatory of delinquency) and to compensate for "a manifest lack of knowledge or expertise" [1] (p.1). Land observatories are no exception and have been multiplying in number over the last 20 years in a context in which decisions concerning land use and land access are becoming increasingly contentious. For instance, the rapid increase in the global

population, the growing complexity of governance (multi-level and multi-actor), and the food price crises of 2008–2009 have led to the growing involvement of international actors and a sudden increase in land demand [3].

The rise of land observatories seems to highlight a growing need for knowledge and information, a need driven by the scale and diversity of land issues. Demands for land observatories depend on the context, which ranges from land reform to large-scale land acquisitions and related land conflicts. As a result, land observatories have been promoted as mechanisms for improving data reliability, reducing information asymmetries, promoting transparency, and thus supporting informed decision-making in favor of citizen participation in land governance [1,4,5]. It is interesting that, in contrast to the initial definitions of Piron (1996) and Jospin (1996), which highlight the State-led and national nature of observatories, the more recent initiatives have diverse structures, are implemented by various types of actors, and are deployed at different levels, ranging from the very local to the global. For instance, the Land Matrix initiative is a global observatory implemented by academics and civil society for monitoring large-scale land transactions, while the Senegal National Land Governance Observatory (ONGF), established by farmer organizations and civil society, monitors and raises awareness on land allocation and dispossession at the local level.

Existing research has failed to acknowledge the diversity of land observatories and their real impacts in the land governance arena. While there exists a significant body of literature on the structural principles of observatories [6–10], there is little research on their conditions of emergence, effective organization, and concrete activities. In the context of land governance, a 'panacea' can refer to a blueprint for a single type of governance system (e.g., government ownership, privatization, community property) or a single type of governance institution [11,12], which is applied to all land problems. The aim of this paper is, thus, to present empirical and analytical evidence to nuance this tendency to consider land observatories as panaceas. In particular, this paper intends to define ways of identifying the diverse nature of land observatories and their determining characteristics. Acknowledging the heterogeneity of land observatories will allow support (whether technical or financial) to be provided for the needs of the specific land observatories.

The diversity of land observatories and their determining factors will be apprehended by, first, identifying all existing and past observatories using a snowball sampling, based on literature and via key informants. Second, the analytical framework will be defined using i) existing literature on land observatories, ii) detailed assessments of land observatories based on semi-open interviews conducted via video conferencing, iii) fieldwork and a visit to an observatory, with interviews being held with numerous actors engaged during a short stay at the observatory (Burkina Faso), and iv) participant observation by direct engagement and work at land observatories (Madagascar, Senegal).

The material and methods will be presented in the next section. In the results section that follows, the analytical framework will first be developed, after which we present a typology of land observatories, along with their conditions of emergence and the factors of divergence. The paper also discusses the diversity of land observatory forms and their conditions of sustainability, as well as the interest generated by their implementation within the land governance arena.

## 2. Materials and Methods

### 2.1. Identifying Case Studies

The choice of Africa as a geographical area of study is justified by the scale and diversity of recent land monitoring initiatives on that continent. Even though land observatory initiatives in Africa have more than 25 years of history (Mali in 1994; Chad in 2001), projects to establish land observatories have proliferated over the past decade (Madagascar, 2007; Burkina Faso and South Africa, 2014; Senegal, 2015; Uganda, 2017; Cameroon, 2019) and reflect the need for acquiring knowledge and information on land issues in Africa. In addition, as land observatories are not unique to sub-Saharan Africa, we are presently also working on expanding our analysis and methods to Asia and Latin America in order to

broaden our material. This cross-continental analysis of land observatories will allow us to apply the conceptual framework detailed in this article to other contexts, and—if necessary—to fine-tune it so as to be globally usable in support of land observatories world-wide. However, the aim of the current article is to develop a framework on a particular set of land observatories on one continent (Africa), to highlight their heterogeneous nature, allowing for expanded analyses to be made in the future.

The first step of the study was to list every institution, irrespective of whether its name included the term "land observatory", in which the main object of study is the monitoring of land dynamics. To do this, an Internet search was conducted for each French-speaking African country with the keywords "observatoire du foncier", "structure foncière" and "outil de suivi-évaluation des dynamiques foncières". For English-speaking countries on the continent, the terms searched for were "land observatory", "land structure", and "land monitoring tool", and, for Portuguese-speaking countries, "observatório terrestre". In addition to the Internet search, discussions were held with resource persons (contacts identified via the land observatories' websites, contacts known to the supervising team, and experts who have written articles/reports on land observatories). Finally, a literature review was carried out, in particular, of documents obtained from land observatories and land structures/institutions, as well as of reports from experts who have analyzed and/or supported the setting up of land observatories. A (non-exhaustive) list of 26 land monitoring initiatives (abandoned, ongoing, and in preparation phase) is provided in Figure 1.

All identified land monitoring initiatives on the African continent were represented on a map (Figure 1). This map highlights two main results. Existing or planned land observatories are located in the four sub-Saharan African subregions: West Africa (Burkina Faso, Senegal), East Africa (Madagascar, Uganda), Central Africa (Cameroon, Chad) and southern Africa (South Africa). No land observatories or similar structures were identified in countries in North Africa, while some countries, such as Senegal or Cameroon, have several each. The lack of available information, data, and contacts, as well as the very embryonic status of some observatory projects (limited discussion between some land stakeholders), hampered the analyses of the following observatories: Observatory on Territorial and Land Dynamics (northern Benin), Civil Society Organization Land Observatory (Benin), Paul Ango Ela Foundation Land Observatory (Cameroon), Observatory of Large-scale Land Acquisitions (Cameroon), Rural Observatory (Mozambique), Congo Land Observatory (Democratic Republic of the Congo), Congo Basin Land Observatory (Democratic Republic of the Congo), Land Observatory of the National Chamber of Kings and Traditional Chiefs (Côte d'Ivoire), Land Observatory of the Primature (Côte d'Ivoire), Observatory of Rural Land (Côte d'Ivoire), Central African Forest Observatory (Member States of the Central African Forest Commission), and the West African Regional Rural Land Observatory (Member States of The West African Economic and Monetary Union (UEMOA)).

Nevertheless, the African focus, together with additional selection criteria (such as i) engagement in observatory-specific functions at the time of the survey (data collection and/or production, monitoring and evaluation of land dynamics, dissemination of analyses, etc.), ii) availability and accessibility of information (in order to be able to undertake initial research—remotely, as no financial means were available to travel all over the African continent, and iii) availability of contact information and response rate (in order to carry out telephone interviews and email exchanges)), resulted in the sample for this article being limited to nine land observatories. The nine selected land observatories are located in Burkina Faso, Cameroon, Chad, Madagascar, Mali, Senegal, South Africa, and Uganda (Table 1). Based on secondary information available about the full set of 26 land observatories, it was evident that the nine observatories retained for in-depth analysis covered the diversity of observatories on the continent. For future work, broadening the sample of land observatories with additional case studies from Africa and from other continents would allow for improved statistical analyses and assessments to be made, strengthening the overall quality of our typological work and validating (or not) the proposed typology.

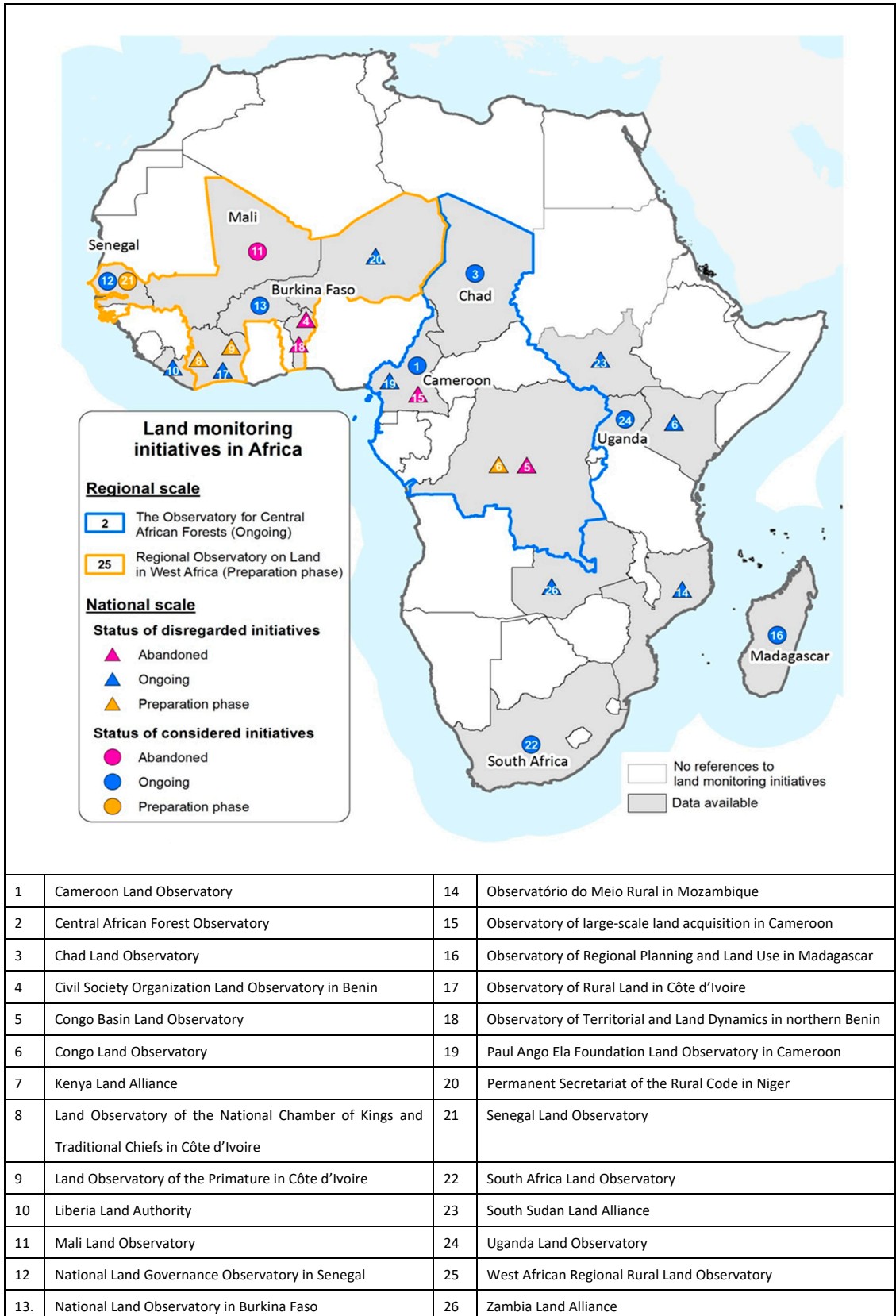

| 1 | Cameroon Land Observatory | 14 | Observatório do Meio Rural in Mozambique |
|---|---|---|---|
| 2 | Central African Forest Observatory | 15 | Observatory of large-scale land acquisition in Cameroon |
| 3 | Chad Land Observatory | 16 | Observatory of Regional Planning and Land Use in Madagascar |
| 4 | Civil Society Organization Land Observatory in Benin | 17 | Observatory of Rural Land in Côte d'Ivoire |
| 5 | Congo Basin Land Observatory | 18 | Observatory of Territorial and Land Dynamics in northern Benin |
| 6 | Congo Land Observatory | 19 | Paul Ango Ela Foundation Land Observatory in Cameroon |
| 7 | Kenya Land Alliance | 20 | Permanent Secretariat of the Rural Code in Niger |
| 8 | Land Observatory of the National Chamber of Kings and Traditional Chiefs in Côte d'Ivoire | 21 | Senegal Land Observatory |
| 9 | Land Observatory of the Primature in Côte d'Ivoire | 22 | South Africa Land Observatory |
| 10 | Liberia Land Authority | 23 | South Sudan Land Alliance |
| 11 | Mali Land Observatory | 24 | Uganda Land Observatory |
| 12 | National Land Governance Observatory in Senegal | 25 | West African Regional Rural Land Observatory |
| 13. | National Land Observatory in Burkina Faso | 26 | Zambia Land Alliance |

**Figure 1.** Land monitoring initiatives in Africa.

**Table 1.** Basic information on nine land observatories based on certain core dimensions (complete table with all core dimensions can be found in Supplementary Materials).

| | Launch Date | Contexts that Prompted the Establishment of the Observatory | Actors Who Supported the Observatory Project | Funding Source |
|---|---|---|---|---|
| Mali | 1994 (abandoned in 1998) | *Monitoring and assessment:* Mismatch between the country's land laws (priority accorded to the State domain) and the inhabitants' land practices | Ministry of Agriculture, Livestock and Environment, and the *Caisse Française de Développement* | *Caisse Française de Développement* |
| Chad | 24 April 2001 | *Land context:* Management of land conflicts | The State via the Ministry of Higher Education, Research and Innovation, hosted by the University of N'Djamena | Ministry of Higher Education, Research and Innovation |
| Madagascar | February 2007 | *Land context:* 2005 land reform | Experts of the National Land Programme and the *Centre de coopération internationale en recherche agronomique pour le développement* (Cirad) who conducted the feasibility study | French Development Agency (AFD) via the implementation of the *Contrat de désendettement et de développement* (C2D) and the International Fund for Agricultural Development (IFAD) |
| Burkina Faso | 3 July 2014 | *Monitoring and assessment:* Monitoring and assessment of the application of Act No. 034-2009/AN of 16 June 2009 on rural land tenure and Act No. 034-2012/AN of 2 July 2012 on agrarian and land reorganization | Millennium Challenge Account—Burkina Faso and CIRAD (which conducted the feasibility study) | Millennium Challenge Account—Burkina Faso |
| South Africa | 1 October 2014 | *Monitoring and assessment:* Land holding monitoring | University of Pretoria (university sphere) | Flemish government |
| Senegal (ONGF) | 17 June 2015 | *Land context:* Democratization of land governance (citizen participation in land management) | Farmer organizations and civil society | European Union Delegation and the Rosa Luxemburg Foundation |
| Uganda | 1 October 2017 | *Monitoring and assessment:* Large-scale land acquisition monitoring | LANDnet Uganda, the Land Matrix Africa and civil society | The Land Matrix |
| Cameroon | January 2019 | *Land context:* Implementation of the national land governance action plan | Civil society organizations and International Land Coalition | International Land Coalition and the Land Matrix |
| Senegal (ONFS) | Planned | *Monitoring and assessment:* Monitoring and assessment of the application of land policy in Senegal | Project spearheaded by many actors: university, research community, State, civil society | Under discussion |

## 2.2. Description of the Data Collection Process

A total of 24 interviews (via video conference or face-to-face during field missions) was conducted with the following organizations and stakeholders: land observatories, Ministries (land, agriculture, and fisheries), civil society organizations (Solidarité des intervenants du foncier in Madagascar, Conseil National de Concertation et de Coopération des Ruraux in Senegal, etc.), consulting firms (INSUCO in Burkina Faso), research and academic entities (Agricultural and Rural Foresight Initiative and the Senegal Agricultural Research Institute in Senegal, Centre de coopération internationale en recherche agronomique pour le développement (Cirad), University of Pretoria, etc.), associations (Association for the Promotion of the Livestock in the Sahel and the Savanna in Burkina Faso), and land experts. In addition, data collection and analysis benefitted from the authors' experience and knowledge (founding members of the Land Matrix, long-term involvement in the land observatory in Madagascar, support for the implementation of a national observatory on large-scale land acquisitions in Senegal, etc.). The interviews, therefore, provided an opportunity to meet a wide range of stakeholders from a variety of backgrounds. Field visits were undertaken in Senegal (three of the authors were based at the Senegalese Institute of Agricultural Research during the study) and Burkina Faso (one-week mission), during which a large number of actors were interviewed (national land observatory, UEMOA Commission, INSUCO, Action and Land Research Group, etc.), as well as in Madagascar. Every entity we contacted as part of the study had some link to land observatories. These links were direct in cases of civil society organizations that were members of the observatory (such as in Senegal or in Uganda), observatories which came under the ambit of a ministry (such as in Madagascar), etc. Links were indirect in cases of a research office or institute that carries out studies on the impact of a land observatory's activities on land governance. As a result, their interest in the dynamics of land observatories and, more generally, on land issues could diverge.

## 2.3. Typological Method

As part of this study, we have chosen to develop a typology of land observatories based on their functions. The typological method, understood here as a descriptive and comparative typology [13], appears to be suitable for representing a diversity of structures and for an organized and intelligible description of concrete and dynamic experiences [14]. Using a continent-wide comparison based on qualitative interviews with the actors involved in land observatories and with the multiple actors involved in land governance processes (ministries and local authorities, civil society organizations, farmer organizations, representatives of customary authorities, donors, etc.), this paper highlights the common denominators across the diversity of observatory structures. That said, a significant pitfall of using the typology could be the oversimplification of results. It is, therefore, necessary to avoid falling into the trap of an excessively normative and restrictive framework for land observatories.

## 3. Results

### 3.1. Defining an Analytical Framework to Better Apprehend the Diversity of Land Observatory Forms

Gaining a better understanding of the diversity of land observatory forms and better apprehending the elements of their sustainability requires defining their core dimensions. On the basis of empirical work, eight dimensions were identified to construct the analytical grid: context of emergence, institutional anchoring, financing methods, topics covered, activities carried out, methods for collecting land-related data, tools for disseminating information, and objectives of the activities carried out. Based on these dimensions, we propose to develop an analytical framework characterizing the functioning of land observatories as land management institutions. Thus, irrespective of its form, every observatory has an aim, whether it be data collection, information dissemination, or knowledge production, and so on. Furthermore, each observatory has attributes that depend on the context of its emergence and which lead to different modes of governance, depending on its activities and aims. Finally, to provide the observatory with the means to meet its aims, a number of conditions are necessary. Conditions are

understood as variables such as tools to collect and disseminate data, and financial means. Hence, following the "grammar of institutions" developed by Crawford and Ostrom [15], we operationalize the attributes, aims and conditions of institutions with the core dimensions of land observatories based on our empirical work. The eight dimensions identified in order to represent the diversity of land observatory forms are presented below.

- Attributes represent organizational variables. Land observatory attributes are:

  ○ The context of emergence is fundamental to understanding the factors that triggered the creation of a land observatory (political, social and land context, person or institution that advocated for its creation, the partners, or types of stakeholders involved). Indeed, in a given context (land reform, land conflict, large-scale land acquisitions), in a given country and at a given moment, several forms of observatory managed by different actors can emerge. Thus, analyzing the conditions of emergence of land observatories allows for a better apprehension of the "incentives and resources for institution building" [16]

  ○ A focus on institutional anchoring allows relationships between organizations to be identified. This variable is important for understanding, on the one hand, the extent to which the mode of governance and connectedness of relationships are related to context, and, on the other, the extent to which the status of the observatory is shaped by this institutional anchoring.

- Aim, as defined by Crawford and Ostrom, concerns the action and/or outcome related to the institution. For land observatories, we have defined three sub-variables:

  ○ Topics covered: This variable is used to assess the relative degree of independence of the observatory in the choice of topics covered, and to identify demands and obligations arising from other institutions. Identifying the main topics covered will also highlight major requirements of data concerning the main land issues affecting the country concerned.

  ○ Activities carried out: This variable helps a pragmatic assessment to be made of the nature of the observatory and the understanding of how it can influence land governance processes.

  ○ Objectives: This variable takes into account the observatory's professed direct impacts (raised awareness, decision support, creation of a forum for debate, etc.) and indirect impacts (other organizations from different sectors collect information, community creation around land, etc.)

- Conditions refer to the functioning of the institution. For land observatories, the functioning pertains to the ways of raising financing and of collecting data, and tools for disseminating information and knowledge.

  ○ Financing methods: Going beyond an analysis of the budget structure and the financial sources (multi- or single-source funding, national and/or international donors), we assess the observatory's financial independence, and thus its sustainability.

  ○ Data: This variable is important for helping determine what methods are used to collect land data. Indeed, because land-related data (land deals concluded, concrete impacts of the implementation of a new land policy, management of a protected area, etc.) is difficult to access, and can be sources of rents and conflicts, we wish to better understand the protocol to gain access to data.

  ○ Tools: Apprehending the sensitivity of land data and knowledge and the way to publish information (what data can be disseminated and how people/institutions will use the information).

We have, thus, defined eight dimensions, corresponding to the sub-variables listed above, to characterize land observatories. This framework was then used to structure semi-structured interviews and served

as a basis for all interviews conducted during the study. It also ensured that the data collection protocol remained the same for all the observatories studied.

*3.2. Attributes of Land Observatories*

The establishment of a land observatory depends on a combination of factors. First, the setting up of a land observatory is triggered by a significant change in the land sector (land reform, large-scale land acquisitions, etc.) or a recurring issue (mismatch between the legal framework and land tenure practices, land conflicts, etc.). In Madagascar, the land observatory was created in February 2007 with the initial objectives of analyzing the progress of land reform (2005), assessing their impacts, and proposing relevant guidelines. In Uganda, the land observatory was set up as a result of large-scale land acquisitions that were largely unregulated and which created substantial conflict and led to the displacement of individuals and communities. For example, the New Forestry Company (NFC) obtained permission from the National Forestry Authority (NFA) to establish a carbon offset plantation in Mubende district but displaced over 22,500 individuals in the process, which, in some cases, led to violence and the destruction of property, crops, and livestock. Thus, the Uganda land observatory collects and disseminates data on successful, planned, and failed land deals in the country. In Mali, it was the mismatch between a legal framework, which accorded preference to the State domain, and land tenure practices that led to the establishment of the land observatory in 1994. Over its four years of operation, the Mali land observatory produced summary reports and recommendations to support the preparation of a national land charter, a local-authorities code, and a map of local and regional land issues and their characteristics.

Second, the creation of a land observatory requires the mobilization of a diversity of actors (public/private, national/international, etc.) involved in the land governance arena. Indeed, significant change in the land sector generates questions and engenders debate, creates injustice, and is a source of conflict. As a result, actors are mobilized to propose solutions and to insert these issues into political agendas. In Uganda, the creation of the land observatory was initiated by civil society with the aim of including multiple sectors and actors so that the platform could be broad-based. The main organizations involved in the establishment of the Ugandan land observatory are LANDnet Uganda, Safer World Uganda, Uganda Land Alliance, Coalition of Pastoralist Civil Society Organization (COPACSO) and Food Rights Alliance (FRA). Yet, different trajectories are possible. Indeed, even though led by civil society, the land observatory in Uganda is a multi-stakeholder observatory. It includes private sector partners (Uganda Chamber for Mines and Petroleum), NGOs (International Accountability Program, Participatory Ecological Land Use Management Uganda (PELUM), Southern and Eastern Africa Trade Information and Negotiations Institute Uganda (SEATINI), Transparency International), the government (Ministry of Lands, Housing and Urban Development), and research organizations (Makerere University). In Senegal, the National Land Governance Observatory (ONGF), launched on 17 June 2015, was founded by civil society and farmer organizations (all members of the *Cadre de Réflexion et d'Action sur le Foncier sur la réforme foncière au Sénégal*) with a view of setting up a militant structure. They, thus, excluded the State and the private sector from the observatory. In South Africa, the proponent of the establishment of the land observatory is the academic sector, including the University of Pretoria and the University of Western Cape. Finally, in Chad, the land observatory is an initiative of the State. By Decree No. 01-215, the Government of Chad, through the Ministry of Higher Education, Scientific Research and Innovation, established the observatory officially in April 2001, hosted by the National Institute of Human Sciences of the University of N'Djamena.

Third, the effective implementation of a land observatory depends on the presence of a diversity of donors who are willing to fund the observatory project. In the majority of cases (eight cases out of nine), the observatories were able to be set up with the support of international funding bodies. The nature of donors is heterogeneous: public development agencies (French Development Agency, Swiss Cooperation), international network of intergovernmental organizations and civil society (International Land Coalition), international organizations (International Fund for Agricultural

Development (IFAD), Food and Agriculture Organization (FAO)), and foreign governments. In Burkina Faso, it was within the framework of the implementation of the "Land tenure security" project of the Millennium Challenge Account—Burkina Faso (MCA-BF) that the National Land Observatory was established in July 2014. From the end of 2014, USAID took over and financially supported the observatory through the Resilience in the Sahel (RISE) project. In South Africa, although the initiative was launched by the academic sector, it was the Government of Flanders (Belgium), the entity funding this land observatory, which was responsible for its establishment. In Senegal, the creation of the national land governance observatory was supported by many partners (Rosa Luxemburg Foundation, European Union, OXFAM via ActionAid, IFAD, FAO, and Rights and Resources Institute (RRI)), which made it possible to organize numerous training and exchange workshops prior to the actual setting up of the observatory.

### 3.3. Aims of Land Observatories

The main observation domains of the land observatories studied are rural land and, to a lesser extent, urban land. Given the diversity of observation domains, an observatory can also cover many different topics. However, the study revealed that the gradual increase in the number of topics covered by an observatory is related to its funding, experience, and legitimacy in the land sector. For instance, the Malagasy land observatory specialized on a specific topic when it was established, i.e., monitoring of land reform. It subsequently dealt with other observation domains and topics as it gained experience and recognition (number of land disputes in the courts, monitoring of large-scale land transactions, access to land for young people, land security and reforestation, etc.). For an observatory to be sustainable, the topics covered have to be of interest to decision-makers and donors in order to attract funding, and the findings must be disseminated by the media and be driven by community demand. Hence, in Uganda, the primary mission of the observatory, which is based on the same rationale as the Malagasy land observatory, is to monitor large-scale land transactions—which is a topical issue for donors (the Land Matrix as an international observatory focused on large-scale land deals), a concern of decision-makers (the Land Matrix website has identified 49 land deals concluded in Uganda, mainly since 2010, representing 277,444 ha) and a challenge for local communities (land grabs trigger population movements and conflicts). In contrast, the National Land Observatory in Burkina Faso (ONF-BF) has come in for criticism for the over-ambitious scope of its five-year observation action plan. The observatory has 14 observation domains that deal with rural, urban, and periurban land. Many observers believe that this plan is too far-reaching and is partly the reason behind the limited number of actual results/studies published on the observatory's website.

In addition, the analysis of the observatories' primary activities showed what distinguishes the observatories from each other. The national observatory of land governance in Senegal—driven by civil society—is an engaged and militant advocacy structure that acts as a whistleblower in order to influence the public debate. The observatory currently does not produce thematic studies, nor does it carry out field surveys. Indeed, ONGF's primary missions are to monitor and issue alerts on land governance and land allocation (mainly via its Facebook account) and to facilitate stakeholder awareness and mobilization. In contrast, as a research structure and generalist land observatory, the Malagasy land observatory provides training, produces primary data, organizes restitution workshops, leads political debates, supervises PhD theses and internships, etc. For its part, the Uganda land observatory is a land monitoring structure of which the task is to collect land data on large-scale land acquisitions across the country from data-holding structures (Ministry of Lands, Housing and Urban Development) and decentralized structures. Indeed, when conducting research on large-scale land acquisitions in the country, this land observatory relies on focal individuals on the ground. The country is divided into 14 blocks, each representing an administrative region comprising a number of districts. Each of these regions is coordinated by a regional researcher who is the focal individual in that region. The regions are: Buganda, Ankole, Acholi, Teso, Busoga, Bunyoro, Toro, Kigezi, Karamoja, Bugisu, Lango, Sebei, Bukedi, and West Nile. The observatory then disseminates data via its website on successful, planned,

and failed large-scale land acquisitions in the country. Thus, each observatory, depending on its aims and mandates, will undertake different activities on different topics with different impacts on land governance.

By focusing on a specific phenomenon (land conflicts, large-scale land acquisitions, the premise of land rights formalization as a trigger of investment behavior, etc.), the observatory not only sheds light on this issue but also helps to establish it as such and to give it a place in the public sphere and on the political, social or scientific agenda [6]. For instance, in Uganda, large-scale land-based investment is steadily on the increase following the creation of a conducive environment by the government through the enactment of the National Land Policy of 2013, which allows for Ugandan and foreign citizens alike to acquire land for investments purpose. Although several organizations have been working on land issues in Uganda (LANDnet Uganda, Uganda Land Alliance, Land and equity Movement Uganda, Participatory Ecological Land Use Management, Ministry of Lands Housing and Urban Development, etc.), none of these entities is monitoring land with a specific focus on large-scale land acquisitions. This issue was introduced and put onto the political and policy agenda by the establishment of the Uganda land observatory.

### 3.4. Conditions of Land Observatories

Data access is an ongoing and persistent problem for land observatories. An observatory's proper functioning and success depends to a large extent on its ability to take into account and find ways to circumvent obstacles to accessing land data. Open-access data provided by State institutions is often not comprehensive since land-related data and analyses are very political and sensitive. This is especially the case in Cameroon, Uganda, and South Africa, where government institutions view land information as a source of power and income (information collected via video conferencing). In such contexts, an observatory often has limited access to data due to prohibitively high costs and political or technological constraints. Access to information may nevertheless be possible, depending on the institutional positioning of the observatory or its network. For example, the Malagasy land observatory, thanks to its institutional positioning—affiliated with a ministry—can more readily access administrative data.

Identifying who observes, acquires, and produces data is essential for gaining an understanding and defining the role of land observatories. Irrespective of whether the observatory is a multi-stakeholder platform or a civil society organization, ad hoc studies, periodic surveys and regular production of monitoring data are undertaken, above all, by a small core group of determined and motivated experts who conduct the studies on behalf of a larger group. At the Malagasy land observatory, for instance, only the director and study leaders contribute to qualitative analyses, although the observatory collaborates with ministry officials, the private sector, national and foreign universities, and foreign research institutes. Observation is generally assigned to a small group of actors (a researcher and an observatory expert) to avoid two pitfalls: (i) spreading observation and activity management to many actors may disperse the responsibilities and in turn, give rise to organizational problems, and (ii) assigning observation to actors outside the observatory may risk undermining the quality of the studies being conducted.

However, not all observatories produce primary data and conduct field studies. Indeed, each observatory develops various land data collection tools, depending on its resources, methodology, and the topics it covers. For example, the South African land observatory does not produce primary data. It collects data from public or private data-holding structures, with which it has entered into contracts or agreements to do so, and uses its website to capitalize and disseminate data pertaining to land ownership monitoring in South Africa. In Uganda, in addition to data collection through data-holding structures, the observatory utilizes the 14 regional cells to transmit information on large-scale land acquisitions from the local to the central level in order to centralize and disseminate data through its website.

Finally, the tools (website, social networks, articles and studies, organization of workshops and debates) used by a land observatory to disseminate the analyses produced are indicative of the observatory's effectiveness and relevance. Two examples are noteworthy in this respect. The Chad land observatory—the oldest active observatory—does not have any tools for disseminating the analyses it produces, nor has it made available any articles or studies. In contrast, the Malagasy land observatory, in addition to its website (http://www.observatoire-foncier.mg/) that disseminates 80% of the observatory's documents (centralization of articles, publications, press releases, cartography, reflections, mission notes, etc.), maintains links with national media outlets. It also produces and disseminates reports and is active on social media platforms. The analyses produced by the observatory are, thus, widely disseminated and readily accessible. The observatory's main objective is to facilitate discussion between institutions and promote a participatory process through the workshops and conferences/debates it organizes or in which it participates.

### 3.5. Towards Identifying Types of Land Observatories

Analyzing land observatories from the perspective of the core dimensions has allowed us to identify the factors of divergence and the diversity of these land monitoring structures in terms of their contexts of emergence, institutional positioning, modes of governance, and manner of data collection. By considering their functions and activities, we were able to design a typology of land observatories and identify four archetypes with different functions.

The first type can be defined as comprising whistleblowers, exemplified by the national observatory of land governance in Senegal. This type of observatory provides its members with visibility and legitimizes their presence in workshops and debates concerning land issues. This whistleblowing role consists of influencing public debate and raising awareness—amongst a group of actors or a community of interest (farmers, civil society organizations, local populations)—of the negative impacts of certain land dynamics, such as problems of farmer evictions triggered by large-scale land acquisitions. An example is the land grabbing in the Senegalese villages of Dodel and Démette in 2017, leading to mobilization against it. Through advocacy (in particular through social networks) and mobilization of civil society and peasant organizations *(Conseil National de Concertation et de Coopération des Ruraux)*, the national observatory of land governance in Senegal gave these cases an important weight in the public sphere and on the political agenda and contributed to the cancellation of the agro-industrial project by President Macky Sall in November 2018.

The study also identified certain observatories as land monitoring structures (Cameroon, South Africa, and Uganda). Land monitoring involves: (i) identifying/searching for land data, (ii) pooling these data using relatively sophisticated technological tools (contracts and agreements signed with land data-holding structures, telephonic exchanges, SMS, hardcopy transmissions), (iii) storing them over the long term (libraries and resource centers), (iv) analyzing them, and (v) distributing them via social networks, websites developed by observatories, and the national media. In summary, the two main land monitoring functions are to facilitate the understanding of data and access to data. The observatory, thus, enters into agreements and/or contracts with entities holding land data. It is not itself involved in primary data production, nor does it carry out field missions and surveys. It collects secondary data through private or public structures that can share land data. This is the case of the South Africa land observatory. It has signed data-sharing agreements with several land data-holding structures, including the Ministry of Rural Development and Land Reform. For this type of observatory, it is essential to set up a website to collect, store, and disseminate land data. The South Africa land observatory website (https://salandobservatory.org/) has been operational since 2015. It allows users to publish information on land use, land capability, land deals, and farm size in South Africa, gain access to the observatory database, and download all documents published on the site, free of charge.

A third type of observatory focuses on monitoring and assessing land policy. An observatory of this type develops observation indicators (quantitative/qualitative, rural/urban) in order to measure the level of progress of land reform and to assess their impacts, as was the case for the land observatory

in Madagascar during the first two years of its existence (2007–2009). The observatory manages an information system based on about 20 indicators, including the number of land tenure offices created and the number of certificates requested and issued. At the same time, it provides qualitative analyses of the institutional process for implementing the land reform. The observatory was created as a tool to promote land governance with the initial aim of analyzing the progress of the land reform and of drawing up relevant guidelines. The Malagasy land observatory was, thus, more of a land-reform observatory than a generalist land observatory during its first two years of operation.

Finally, the last type of observatory we have identified is the multipurpose observatory. On the one hand, an observatory of this type collects, stores and manages data, and, on the other, it analyzes the data to be able to generate knowledge and information, which it subsequently disseminates through reports. Primary data production is the main characteristic that distinguishes this type of observatory from the other identified forms. Such is the case of the Burkina Faso and Malagasy land observatories. The former has conducted six thematic studies since its launch in 2014, spanning the country's seven social-land zones on the formalization of land rights, knowledge of land policies and laws, land conflicts, and spatial planning instruments (summaries are available on the ONF-BF website). Furthermore, every three months, the observatory publishes a paper document (*Zoom sur le Foncier*) that covers current land issues in Burkina Faso. It also participates in international conferences, which helps it to develop its network and gain visibility, and hosts students on internships.

## 4. Discussion

### 4.1. Looking Beyond the Diversity of Land Observatories

This paper's aim was to analyze a pool of land observatories and identify different types of them. Our results highlight the diversity of land observatories and their structural dimensions. We show that, at a minimum, all land observatories studied share common traits: they collect and disseminate data, and they are established in response to significant land issues and when external donors are willing to fund them.

Despite these similarities, the different observatory types concern different roles and mandates, and are also linked to very specific contexts and stakeholders.

We show that these differences cannot be tied to singular dimensions of land observatories but to combined specifications. The need for considering multiple vantage points explains why, in a given country, at a given moment and in a similar land context, various land observatories initiatives can emerge, with disparate missions and aims, carried forward by diverse actors.

For instance, several forms of land observatories emerged in a similar context in Côte d'Ivoire, where a feasibility study for a national land rural observatory was conducted between August and November 2018 at the request of the State Prime Minister's office (Primature). The State's intentions were to create a decision-making tool to forecast future issues that might result from the recent land reform. At the same time, a civil society organization associated with the National Chamber of Traditional Kings and Chiefs was setting up an observatory (without, however, calling it as such) that would use the grassroots and nationwide network of traditional kings and chiefs to track all information pertaining to land conflicts and dysfunctions arising directly or indirectly from the implementation of the new land reform.

Similarly, we stress the key importance of apprehending land observatories through their multiple dimensions in order to avoid the pitfalls of simple one-size-fits-all solutions. We, thus, acknowledge that the importance of institutional arrangements need to be put into perspective as one can be successful in a particular setting and fail in another. This is illustrated by the differences between the dynamic government-led Malagasy land observatory and the low-impact, multi-stakeholder observatory in Burkina Faso, as well as by the differences between the productive multi-stakeholder Ugandan observatory, and the long-established but largely inactive observatory in Chad. This acknowledgment allows us to temper illusions of an institutional panacea [11,12,17–19].

In addition to the diversity of land observatory forms, we should also note that these different forms are not static, nor are they destined to remain frozen over time. Observatories, like the objects they analyze (land reform, large-scale land acquisitions, land titles issued by land administrations, etc.), are dynamic tools, which adapt and evolve according to contexts and periods. Thus, the Malagasy land observatory has known two phases in its evolutionary dynamics. The first lasted from 2007 to 2009, when the observatory was described as a tool for the monitoring and assessment of the land reform. The observatory's technical structure, attached to the National Land Programme (NLP), consisted of two executives, and its primary mission was to monitor and evaluate land decentralization. The second phase, which began in 2010, has seen the observatory expanding its ambit from a limited monitoring and evaluation function to embracing various analytical themes, diversifying its methods of capitalization and communication, and expanding its staff. This gradual empowerment has helped the observatory become a key player in land debates in the country.

*4.2. The Sustainability of a Land Observatory: A Major Issue for Discussion*

Since land observatories address dynamic phenomena, which evolve over time (analyzing the progress of land reform, monitoring the ownership and size of properties, assessing large-scale land acquisitions, etc.), there are many who advocate for their durability. Our analysis, however, shows that the sustainability of an observatory over time should not be seen as an essential factor in judging its quality. Setting up an observatory for a specific mission with a predefined budget in a limited time frame with concrete results and sizeable impacts seems preferable to designing a long-term observatory to monitor the evolution of land issues without developing strategies for it to have an impact during its various phases through knowledge production and information dissemination. An apt illustration is the Chad land observatory, established by decree in April 2001 and hosted by the National Institute of Human Sciences at the University of N'Djamena. This observatory is overseen and funded by the Chadian Ministry of Higher Education, Research and Innovation (MESRI). From 2006 to 2009, it received additional financial support from the World Bank through the Local Development Support Project. However, as far as its activities are concerned, apart from the publication of a scientific journal entitled Cahier du foncier au Tchad in 2010, the observatory has not undertaken other studies, nor has it brought out other publications. The oldest active observatory in Africa does not seem to have the tools or the means for disseminating information and influencing policy.

Looking beyond a land observatory's sustainability, we argue that its "success" could also be dependent on its legitimacy. Indeed, as a tool for disseminating information and knowledge, the success of an observatory, considered as a knowledge system, must be put into perspective with the salience, credibility, and legitimacy of the information it produces [20,21]. For instance, the Malagasy land observatory is considered a reference in Africa because of the salience of the studies carried out, the credibility of the analyses produced, and the legitimacy acquired by the observatory with national stakeholders (ministries, civil society) and international ones (donors). In this way, the observatory has managed to continue to function effectively for over 10 years, despite political and economic crises experienced in the 2009–2014 period.

## 5. Conclusions

In this paper, our objective was to analyze the diversity of land observatory forms (involving various stakeholders, providing different functions according to the context of the establishment, integrating various methodologies, etc.). Through a study encompassing nine observatories in Africa, we developed an analytical framework based on eight dimensions structuring these institutions. Four types of land observatories were identified on the basis of their functions. In addition to arriving at these innovative results, we also argue that this analytical framework is a practical way for stakeholders to design the pathway towards the land observatory they want to establish and to better apprehend the elements that will ensure its sustainability, as well as for an improved understanding by external actors of the nature of land observatories. The concrete impacts of land observatories in land

governance, power relations between actors (within and outside the observatories), and the strategies implemented by observatories to access and share land-related information, will have to be further analyzed.

**Supplementary Materials:** The following are available online at http://www.mdpi.com/2073-445X/9/3/70/s1.

**Author Contributions:** Conceptualization, Q.G., J.B., W.A., P.B., and E.H.; Methodology, Q.G., J.B., W.A., P.B., and E.H.; Validation, Q.G., J.B., W.A., P.B., E.H. and D.D.; Formal Analysis, Q.G., J.B., W.A., and P.B.; Investigation, Q.G.; Writing-Original Draft Preparation, Q.G.; Writing-Review & Editing, Q.G., J.B., W.A., and P.B.; Supervision, J.B., W.A., and P.B. All authors have read and agreed to the published version of the manuscript.

**Funding:** This research was funded by the Land Matrix Initiative.

**Acknowledgments:** The authors thank the observatories' focal persons and coordinators interviewed at various stages of this research project. Their experience and knowledge were of great value. The research on which this paper is based was supported by the Land Matrix Initiative, Cirad, and the International Land Coalition. We also wish to thank the macroeconomic analysis office of the Senegal Agricultural Research Institute (ISRA BAME) in Dakar for providing a supportive research environment. The authors also thank Kim Agrawal for the editing of the document.

**Conflicts of Interest:** The authors declare no conflict of interest.

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
