# Peer review of "Going Beyond Panaceas: The Diversity of Land Observatory Forms in Africa"

_land, doi:10.3390/land9030070_

Round 1

Reviewer 1 Report

This paper describes an interesting set of case studies that will be of interest to readers of the journal, although more development is needed for the material to achieve a general relevance.  The material is well-written and presented.

A population of 9 individual Observatories is small for understanding the 8 dimensions used in the analytical grid.  Additionally, all the case studies are from Africa.  I suggest that addressing three issues might focus the paper and help to set the case studies within a wider context.

 How does the concept of the Land Observatories (in Africa) relate to similar practises (either institutions or functions) in other areas of the World?  Setting the African examples within a wider, more global, context, would help to establish the relevance of the case studies and this paper in a broader context.  Are land observatories unique to sub-Saharan Africa? Do the functions of land observatories exist within other institutions in other parts of the world?  Perhaps Government departments or research institutions carry out these functions in Europe or N America?  Can the authors provide this wider context, as I think it would be helpful.  Given this, the paper could also usefully discuss the roles and relevance of Land Observatories as an Institution, within the general context of governance and institutions as they influence land use, land change, and land systems.  Institutions and governance are included as general drivers of change in land systems - what influence can and do the observatories have?  The discussion of sustainability of land observatories, and the way that so much of their funding is from international donors (section 4.3), could be expanded, not only in context of institutions and governance (point 2 in this list of issues), but also in the context of international influences on national land policy and use, and, especially, given that the case studies are in sub-Saharan Africa, on land use issues such as international development, land grabbing, and social/environmental justice.  

I hope that the authors find these suggestions helpful.  

Author Response

Dear reviewer,

Thank you very much for your revisions and comments on our article “Going beyond panaceas: the diversity of land observatory forms in Africa” (Manuscript ID: land-714539). Your comments are of great value and allowed us to improve the quality and accuracy of the article.

Please find attached our responses and propositions.

Thank you in advance,

Sincerely

Reviewer 2 Report

This is an interesting and highly relevant manuscript. The proliferation of land observatories in Africa calls for more in-depth analyses of their characteristics and impacts. This paper does an excellent job in providing a useful and well thought-out analytical framework and applies it aptly to nine land observatories.

I have only a few minor comments on (1) how to better present the framework and (2) a some issues related to terminology and phrasing.

On the analytical framework: there is merit to the framework, it is comprehensive and the comparative case study demonstrates its practical applicability. However, I think it would be better to present it in table format rather than in the current bullet point style. This refers to pages 6-7.

Issues around terminology:

There is no such thing as "participatory observation". The correct term is "participant observation". Please correct this in the abstract on p. 1, line 32 and on p. 2, line 80.  

Issues around phrasing:

p. 2, line 74: it's not clear what "their needs" refer to. I reckon it's about stakeholders' needs? Hence, I suggest rephrasing: "to be better tailored to the needs of various stakeholder groups" or similar.

p. 3, line 95: "are old" is not very precise since 'old' is relative. I suggest rephrasing: "have more than 25 years of history" or similar.

p. 5, lines 162-165: one-sentence paragraphs are not good academic practice. Suggest merging this sentence with the previous paragraphs or adding at least another sentence.

p. 10, line 299: I'm not a fan of sentences starting with "but", as it sounds a little journalistic. Maybe replace by "Yet" or rephrase the entire sentence.

p. 10, line 315: suggest rewording "thanks to" (a little too casual) into "with" or "with support from international funding bodies".

p. 11, line 338: nothing "happens naturally" in socio-political fields, hence I would rephrase this part of the sentence.

p. 11, line 347: not sure why "(Land Matrix)" is in brackets there, it would probably need a few more words for better clarity".

p. 12, line 384: suggest removing the brackets and writing instead: "due to prohibitively high costs, political constraints or other factors". 

p. 13, lines 432-434: suggest providing a little bit more context to this case. It's not clear how the case was supported/publicized by the land observatory.

Author Response

Dear reviewer,

Thank you very much for your revisions and comments on our article “Going beyond panaceas: the diversity of land observatory forms in Africa” (Manuscript ID: land-714539). Your comments are of great value and allowed us to improve the quality and accuracy of the article.

Please find attached our answers and propositions.

Thank you in advance,

Sincerely

Reviewer 3 Report

The study presents an overview and a typology of land observatory in Africa. The draft is very clear and very well written in all its parts. I believe it is a valuable collection of information to be used by researchers and practitioners in the field. I recommend publication after addressing some minor issues:

I feel that the sub-heading in the Methods are not necessary I would introduce the Crawford and Ostrom grammar of institutions in the Methods Towards the end of the draft the authors use the word "laboratories" referring to the observatories. Please be consistent throughout the manuscript. line 147, please correct as "firms".

Author Response

Dear reviewer,

Thank you very much for your revisions and comments on our article “Going beyond panaceas: the diversity of land observatory forms in Africa” (Manuscript ID: land-714539). Your comments are of great value and allowed us to improve the quality and accuracy of the article.

Please find below our answers and propositions.

Thank you in advance,

Sincerely

Round 2

Reviewer 1 Report

Thank you for responding to my questions and observations.